# Atomic mechanism of near threshold fatigue crack growth in vacuum

Mingjie Zhao [1,3], Wenjia Gu[1,3] & Derek H. Warner[1,2✉]

Structural failures resulting from prolonged low-amplitude loading are particularly problematic. Over the past century a succession of mechanisms have been hypothesized, as experimental validation has remained out of reach. Here we show by atomistic modeling that sustained fatigue crack growth in vacuum requires emitted dislocations to change slip planes prior to their reabsorption into the crack on the opposite side of the loading cycle. By harnessing a new implementation of a concurrent multiscale method we (1) assess the validity of long-hypothesized material separation mechanisms thought to control near-threshold fatigue crack growth in vacuum, and (2) reconcile reports of crack growth in atomistic simulations at loading amplitudes below experimental crack growth thresholds. Our results provide a mechanistic foundation to relate fatigue crack growth tendency to fundamental material properties, e.g. stacking fault energies and elastic moduli, opening the door for improved prognosis and the design of novel fatigue resistance alloys.

---

[1] Cornell Fracture Group, School of Civil and Environmental Engineering, Cornell University, Ithaca, NY 14853, USA. [2] Laboratory for Multiscale Mechanics Modeling, École Polytechnique Fédérale de Lausanne, 1015 Lausanne, Switzerland. [3]These authors contributed equally: Mingjie Zhao, Wenjia Gu.
✉email: derek.warner@cornell.edu

The growth of cracks governs engineering decisions across a broad range of industry. Yet, in many technologically relevant regimes, the process by which cracks grow remains unknown. A prime example is near-threshold fatigue crack growth in vacuum environments, which can govern failures initiated from subsurface material defects[1,2]. In this case, crack growth per loading cycle can be on the order of angstroms, inhibiting direct observation of the material separation process[3–5]. Accordingly, atomistic modeling may be the best available tool to better understand the phenomenon.

While a large atomistic modeling literature exists[6–10], the link between modeled cracks and laboratory behavior is complicated by disconnects in stress state, simulation geometry, and thermal activation. In cases where care has been devoted to modeling a well-developed cyclic stress field[11–17], fatigue crack growth has been reported. However, such modeling outcomes occur at loading amplitudes well below the corresponding experimentally observed thresholds for fatigue crack growth in vacuum.[18]

The cyclic loads associated with experimentally observed fatigue crack growth thresholds entail micrometers of deformation at the crack tip[19,20]. This well surpasses modern atomistic modeling capabilities, which at best could simulate a sufficient 2D domain for 10's of loading cycles (with a dedicated ExaFLOP supercomputer using ~ $500,000 of electricity). This challenge has limited the focus of previous studies towards nanometer cracks, lower loading amplitudes, and few loading cycles[11–17,21–27]. A concurrent multiscale approach[28] addresses the challenge by reducing degrees of freedom, allowing larger simulation domains to be studied over more cycles without sacrificing atomic resolution at the crack tip. For crack behavior, the multiscaling is complicated by the movement of dislocations over large distances, and thus, a coupled atomistic discrete dislocation (CADD) approach is necessary[29].

Here, we report on atomistic simulations to cycle counts far beyond those analyzed previously by harnessing contemporary computational resources and a parallel implementation of the CADD concurrent multiscale approach. Our simulations show that fatigue crack growth arrests after an initial transient period, reconciling the standing discrepancy between model and experiment. With this understanding, we then examine hypothesized mechanisms for near threshold fatigue crack growth in vacuum[19,30–33]. We find sustained crack growth to only occur when edge dislocations return to the crack tip on a slip plane behind the one on which they were emitted.

## Results

**High cycle count simulations and experimental validation.** We begin by presenting the results of a cyclically loaded mode I crack in an initially dislocation free fcc aluminum crystal. The crack is placed on a (−1 1 3) plane with its front along the [2 −1 1] direction, with details given in the "Methods" section. This orientation is favorable relative to other options, as it enables the study of full dislocation emission in the athermal limit in a thin simulation cell[34]. These two features are necessary to make the simulation of many loading cycles computationally tractable.

An additional strategy to make the many cycle simulations tractable is to limit the size of the plastic zone, reducing the need to integrate the motion of many dislocations over long distances each load cycle. Towards this goal, we investigate the role of dislocation glide resistance in the continuum domain, whereby greater glide resistances reduce the plastic zone size and make the simulations less computationally demanding.

Simulations performed across a wide range of glide resistances, 150–654 MPa, show that fatigue crack growth is independent of this parameter (Fig. 1a). This finding is in accord with discrete

dislocation continuum modeling[35,36] and experiments[2,35,37–41] that show near threshold fatigue crack growth to be independent of dislocation glide resistance and plastic zone size, when crack closure is accounted for or does not occur (at high $R$ values or in vacuum). This result and the supporting literature provide validity for simulations with dislocation motion constrained to the atomistic domain (equivalent to imposing a high glide resistance) to attain high cycle counts.

Simulating beyond the cycle counts previously accessed[11–17,21–27] reveals crack arrest after the first 10's of cycles, independent of the dislocation glide resistance in the continuum domain (Fig. 1b). In the arrested state, the crack tip, defect, and dislocation structure is reversible over the course of the cycle, even with repeated dislocation emission and absorption in some cases.

A challenge to interpreting these results relative to corresponding laboratory experiments is that the modeled crack-crystal orientation and thin periodic cell constrain dislocation slip to a single plane. This challenge cannot be directly addressed, as (1) choosing a multi-slip orientation would lead to crack tip twin emission at atomistic modeling timescales[34,42] and (2) thickening the model to permit slip on other planes would drastically increase the degrees of freedom making high cycle count simulations intractable. Accordingly, we examine an opposite bound to the thin fcc simulations, where dislocation slip is underconstrained. This case consists of a ductile 2D hexagonal lattice with edge dislocation slip on three planes and glissile dislocation reactions. Details are given in the "Methods" section.

Examining a series of loads, the 2D hexagonal lattice exhibits the same response as the previously presented fcc case, with cracks arresting after an initial growth stage (Fig. 1c). Again, this response is found to be independent of the dislocation glide resistance in the continuum, further establishing that emitted dislocations can be constrained in a sufficiently sized atomistic domain, without artificially influencing the characteristics of crack growth behavior. In other terms, we can assume that the plastic zone size does not influence near threshold fatigue crack growth behavior, which is consistent with both experiment[2,35,37–41] and discrete dislocation modeling[35,36] that show yield strength to not influence the fatigue crack growth behavior, when crack closure is accounted for or does not occur (at high $R$ values or in vacuum).

Interpreting the above results to suggest that 2D hexagonal lattice simulations with dislocations constrained to the atomistic domain can be a valid tool for modeling fatigue crack behavior at attainable computational costs, such simulations were performed at higher loads and long cycle counts. As expected, increased loading amplitude led to increased crack growth, with some instances exhibiting sustained crack growth over the entire duration of the simulation, up to 180 cycles (Fig. 1e). Comparison of the arrested and growing cracks highlights the complexity of the phenomenon. In both cases, instances of shielding and antishielding dislocation emission, absorption, and dislocation annihilation can be observed during loading cycles. This suggests that none of these mechanisms are sufficient for sustained near threshold fatigue crack growth in vacuum. The finding that dislocation emission from the crack tip is not sufficient for sustained crack growth in vacuum is consistent with the under prediction of the threshold amplitude by discrete dislocation models governed by this mechanism[43–45].

The occurrence of cyclically reversible/arrested crack configurations does decrease with increased loading amplitude. This suggests that crack arrest is linked to the number of inelastic mechanisms that occur per cycle (such as dislocation emission, absorption, and dislocation reactions). Considering the probabilistic nature of crack arrest and that the modeled loading amplitudes are still significantly below experimental thresholds, we hypothesize that all cracks in Fig. 1e will eventually arrest.

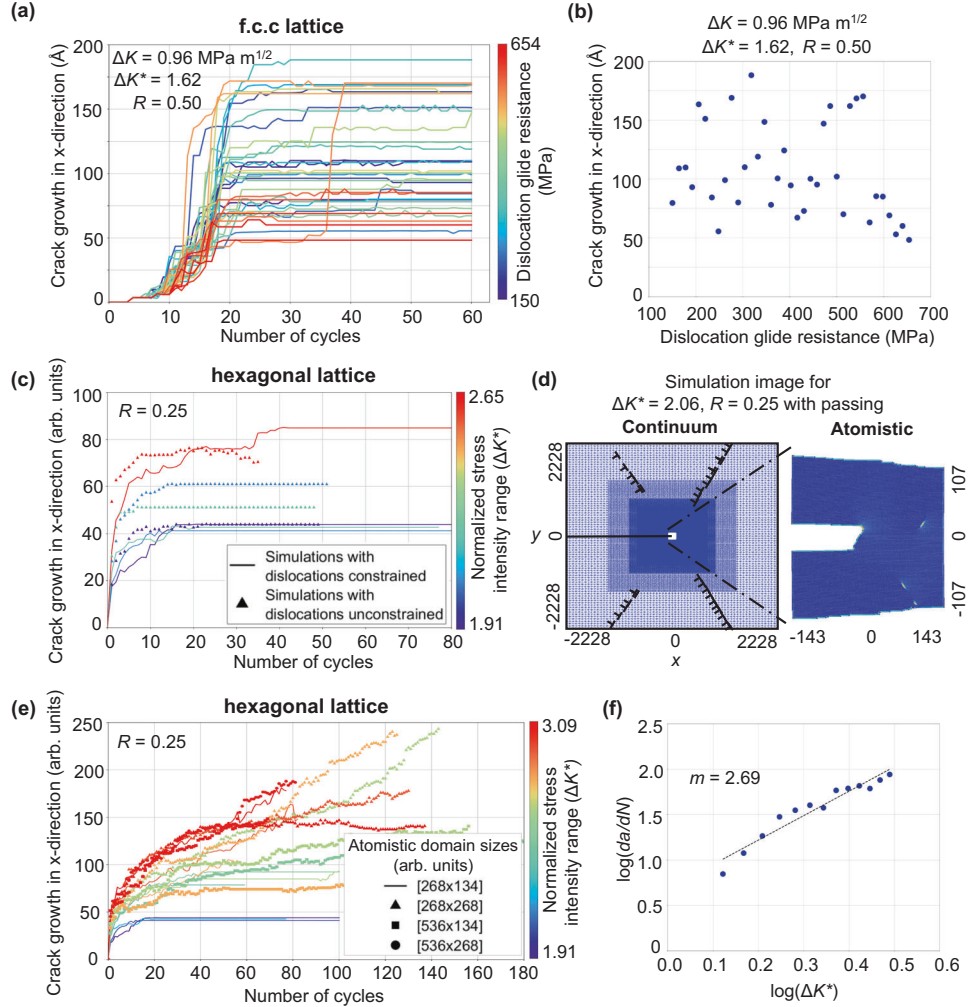

**Fig. 1 Crack advance as a function of loading cycle from three sets of simulations. a** (−1 1 3) crack in an fcc aluminum lattice. Each curve represents a distinct dislocation glide resistance in the continuum domain. The majority of cases show crack growth over the first 20 cycles and crack arrest shortly after. **b** The location of the arrested cracks are shown as a function of glide resistance, showing no correlation between the two variables. **c** A ductile 2D hexagonal lattice at various loading amplitudes. Again, cracks arrest after an initial transient period. Cases are shown both with and without dislocations being constrained to the atomistic domain to establish that restricting dislocation motion does not influence behavior (see Supplementary Fig. 1). **d** The dislocation distribution and the crack tip configuration for an arrested crack, with edge dislocations having nucleated from the crack tip along the two slip planes 60° from the horizontal axis as shown. **e** A ductile 2D hexagonal lattice with dislocations constrained in the atomistic domain at larger cycle counts and higher loading amplitudes. Dimension of the atomistic window size used in each simulation is labeled in square brackets as specified in the legend. Crack arrest was observed in all but a few cases at the higher loading amplitudes. **f** Plot of $\log(da/dN)$ vs. $\log(\Delta K)$ from the first 10 cycles showing the data conforms to Paris' law, consistent with previous atomistic simulations that analyzed crack growth in the initial transient period. $m$ represents the Paris law exponent, i.e. log–log slope. For the 2D hexagonal lattice simulations, distances are normalized by the magnitude of the Burgers vector, $b$, and the range of the normalized stress intensity factor that is used to characterize the loading amplitude, $\Delta K^* = \Delta K_I / K_I^{nuc}$.

We hypothesize that the observation of the crack arrest behavior in Fig. 1 cannot be attributed to our selection of model parameters, but instead represents a more general behavior that has just now been observed by simulating to cycle counts well beyond those examined previously. To support this assertion, we examine the correlation between the rate of crack growth, $\log(da/dN)$, verse loading amplitude, $\log(\Delta K)$, as commonly done in the fatigue literature (Fig. 1f). Extracting data from the initial transient growth region during the first 10 loading cycles, the observed slope (Paris law exponent) is consistent with the atomistic modeling literature limited to low cycle counts. Specifically, a Paris law exponent of 2.7 has been observed in our simulations and 2.8 by Zhou el al.[16], 2.4 by Uhnakova et al.[11], 3.5 by Baker and Warner[13].

**A mechanism for sustained fatigue crack growth in vacuum.** Beyond the benefit of computational cost, the CADD model

provides a means to directly assess the feasibility of previously suggested crack growth mechanisms without requiring increased loading amplitude, which would make simulation to high cycle counts intractable.

In this spirit, we examine the role of plastic slip reversibility, which has been considered key to fatigue failure for over a century[46–51]. Toward this goal, simulations were performed with dislocation glide in the continuum domain being constrained to a single direction, away from the crack tip. As shown in Fig. 2a, cracks quickly arrested in these simulations, despite shielding and antishielding dislocations being continually emitted. Elementary analysis gives no expectation that this result would change at higher loading amplitudes, i.e. slip irreversibility (as implemented) is not sufficient for near threshold crack growth in vacuum.

A less severe plastic irreversibility might involve the formation of debris in front of the crack tip via pinned dislocation segments

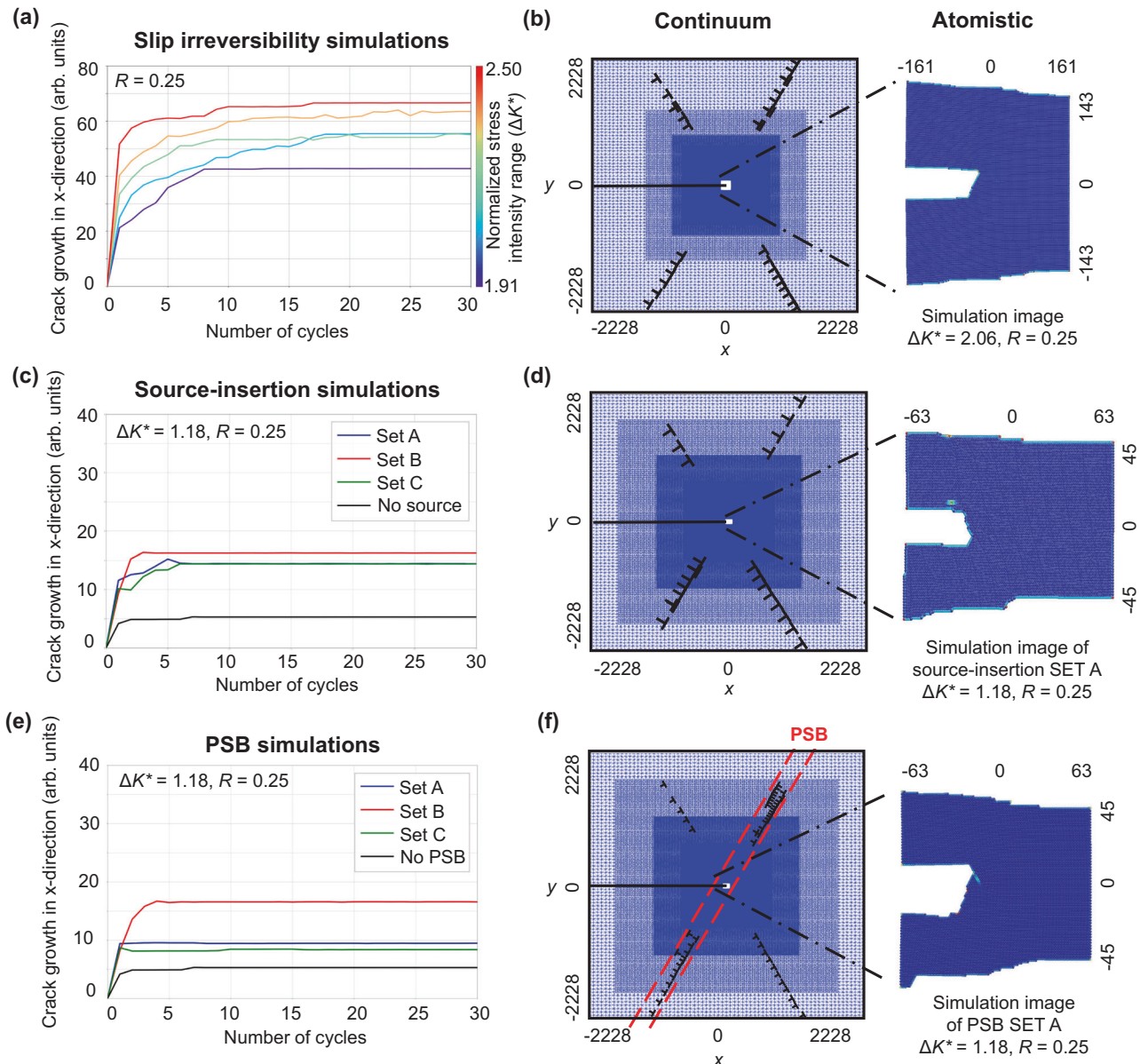

**Fig. 2 Crack advance as a function of loading cycle from three sets of simulations.** Sample continuum and atomistic configurations are shown from when the crack reaches an arrested state. **a** and **b** Simulations with slip irreversibility, where dislocation motion is limited to the direction away from the crack tip. **c** and **d** Simulations with dislocation sources in the continuum, which exhibit enhanced crack growth over the initial cycles, but eventually crack arrest. **e** and **f** Simulations with pre-existing dislocations in the continuum aligned as a persistent slip band (PSB) in red dashed line. The presence of the PSB dislocations enhances growth over the initial cycles, but eventually crack arrest occurs. Details about SET A, B, and C in (**b**) and (**c**) are discussed in the "Methods" section. Distances are normalized by the magnitude of the Burgers vector, $b$, and the range of the normalized stress intensity factor that is used to characterize the loading amplitude, $\Delta K^* = \Delta K_I / K_I^{nuc}$.

and/or junctions that are unable to return to the crack upon unloading. The formation of such debris is not uncommon in atomistic simulations, where it has produced transient periods of fatigue crack growth in some cases[13,15]. Yet, the formation and stability of such debris in atomistic simulations is exaggerated due to the lack of thermal activation[52]. Experimentally, conflicting reports exist, with debris being observed near the crack tip[4] and the absence of it[3], albeit at a coarser scale.

Other suggested mechanisms involve the motion of dislocations into the vicinity of the crack tip. The arrival of such dislocations can provoke cleavage and dislocation emission[35,53] and might be important for sustained fatigue crack growth. To examine this point, simulations were performed with starting configurations having arrays of dislocations and dislocation

sources in the continuum. After preliminary analysis, simulations involving random configurations of dislocations and dislocation sources were abandoned, as this rarely led to dislocations arriving within a range of influence of the crack tip. Accordingly, we pursued simulations involving strategically chosen configurations that would best facilitate sustained crack growth.

Figure 2b shows an example where sources were aligned to emit dislocations towards the crack tip to promote growth. Details of simulation set up is discussed in the "Methods" section. While the presence of such sources does promote crack growth over the first ~5 cycles, the cracks ultimately arrested. Their arrest is due to the stress field of emitted dislocations eventually shielding sources from activating on subsequent cycles such that no new dislocation dipoles are emitted, leading to a reversible configuration.

In a further attempt to produce sustained near threshold fatigue crack growth, preexisting dislocations were aligned in the form of a dense persistent slip band (PSB) configured to send antishielding dislocations to the crack tip, promoting crack growth. Details of simulation set up are discussed in the "Methods" section. The PSB configuration is particularly relevant given the many experimental observations of fatigue crack growth along PSBs[2,54]. As shown in Fig. 2c, the presence of the PSB does enhance growth over the first few cycles. However, the crack eventually arrests due to the formation of stable dislocation arrangements, preventing continued movement of PSB dislocations towards the crack tip to continually facilitate growth. Thus, preexisting dislocations are not a sufficient ingredient for sustaining near threshold fatigue crack growth, at least for the loads and configurations simulated here. Nonetheless, the results do not suggest a lack of interaction between propagating cracks and PSBs, and thus do not contradict the common observations of fatigue cracks at PSBs[55].

Both the simulations with strategically placed sources and PSBs suggest that crack growth could be sustained if dislocations of proper sign continuously arrived at the crack tip as it propagates. The most conceivable means that this might occur is by dislocations emitted from the crack tip changing planes prior to their return on the opposite side of the loading cycle. Such a process would provide a continuously moving source of dislocations that would enable continued fatigue crack propagation via the removal of material at the crack tip.

This hypothesis can easily be tested with the CADD simulation framework by changing the slip plane of emitted dislocations prior to their return to the crack tip. Simulations were conducted at low loading magnitudes where only 1 or 2 dislocations were emitted and absorbed during a loading cycle. Details of the simulations are given in the "Methods" section. Without changing the slip plane of these dislocations, the behavior of the crack at these low loading amplitudes is completely reversible from the first loading cycle. However, when the position of the emitted dislocations changes to a different plane prior to their absorption upon unload, continued crack growth occurs for the case of the new slip plane being behind the crack tip (Fig. 3b). Conversely, when emitted shielding dislocations change planes in the other direction, moving in front of the crack tip, the crack closes.

This behavior can be understood by considering the cyclic motion of two intersecting slip traces, which can remove or place material at the crack tip over the course of a loading cycle (Fig. 3a). The removal of material by the action of intersecting slip traces was hypothesized over 50 years ago by Cottrell and Hull[56,57] for the formation of surface intrusions; and now here, we find its occurrence at crack tips in the simulations presented in Fig. 1e that exhibit fatigue crack growth at high cycle counts (Fig. 4). The movement of material away from the crack tip during cyclic loading inhibits the rewelding of crack faces at the bottom of the load cycle, consistent with carefully executed laboratory observations[58,59].

The intersection of slip traces may be exaggerated in the 2D hexagonal lattice simulations presented here due to the prevalence of glissle dislocation reactions relative to real 3D cubic lattices. Nonetheless, real 3D cubic lattices offer a double cross-slip mechanism that could produce a change in the plane of edge dislocations over the course of a loading cycle[56,57,60]. It is important to emphasize that in either case the dislocation shifting mechanism does not require diffusion, consistent with the weak influence of temperature on fatigue crack growth in the absence of environmental factors[61,62]. Further, the near continuous advance of the crack shown in Fig. 3b should not necessarily be expected, as slip plane shifting might occur in bursts when it results from dislocation processes in real 3D crystals.

In closing, performing fatigue simulations to cycle counts beyond those previously accessible reveals a regime of transient sub-threshold fatigue crack growth in vacuum. This behavior reconciles the reports of crack growth in atomistic simulations at loading amplitudes below experimental crack growth thresholds. Neither slip irreversibility, preexisting dislocation sources, nor preexisting dislocations in the form of a slip band are observed to be sufficient for sustained fatigue crack growth at low loading amplitudes. Sustained fatigue crack growth is only observed when emitted dislocations change slip planes prior to absorbing back into the crack on the opposite side of the loading cycle. This process occurs naturally in simulations by the intersection of slip traces, and is expected to occur in 3D crystals by an additional mechanism, i.e. the long-ago proposed double cross slip mechanism. An easy path towards experimental confirmation is not obvious, but perhaps tracking individual dislocation slip traces at the crack tip via a marked surface and scanning probe microscopy would be illuminating. Future atomistic simulations can harness this finding and that of the transient sub-threshold growth regime to better connect to the actual near-threshold fatigue crack growth process. Such simulations are needed to illuminate the mechanisms controlling environmental, mixed-mode, variable and reversed ($R < 0$) loading effects; which must be incorporated into real-world fatigue prediction models. Ultimately, the slip plane shifting mechanism (whether due to slip trace intersection, double cross slip, or a yet to be determined mechanism) can be expressed in terms of fundamental material properties that can be computed from first principles via Khon–Sham density functional theory. This link between alloy composition and fatigue crack growth resistance provides a route forward towards more robust continuum prognosis models and the design of fatigue resistant alloys.

## Methods

**Simulation framework**. The CADD simulation setup is shown in Fig. 5. We utilized LAMMPS[63] to conduct molecular statics simulations for the atomistic domain and FEniCS[64] for finite-element analysis of the continuum domain. We refer to the method as LF-CADD. The coupling methodology and the discrete dislocation methodology follow Shilkrot et al. [28,29]. The integration of the two open-source packages enabled us to simulate fatigue crack growth more efficiently and in parallel via domain decomposition.

The atomistic domain is governed by an empirical potential and the continuum domain is governed by elastic constants chosen to match the empirical potential. The motion of a discrete dislocation in the continuum domain occurs when the driving force acting on a dislocation exceeds the prescribed lattice resistance. The examined values of lattice resistance were greater than the lattice resistance that occurs naturally in the atomistic domain, creating a spatial discontinuity of glide resistance in the model. This discontinuity is inconsequential for the results presented here as the driving force on dislocations in the atomistic domain is well above the glide resistances that were examined and that of typical engineering alloys.

The crack was created by removing three consecutive planes of atoms. Loads were applied by prescribed displacements at the outer boundary of the continuum domain corresponding to the solution for a sharp crack in an anisotropic linear elastic material subjected to mode I loading of a prescribed mode I stress intensity factor, $K_I$. The linear elastic solution was calculated using an updated position of the crack tip, which was identified using the flood fill algorithm. If the current crack tip position is not tracked and used in the computation of the applied displacements, the stress intensity factor acting on the crack tip will evolve with crack growth.

Cyclic loading was applied by linearly varying $K_I$ between a minimum and maximum value, $K_I^{min}$ and $K_I^{max}$, respectively. The cyclic loading is characterized by the range of the stress intensity factor,

$$\Delta K_I = K_I^{max} - K_I^{min}, \qquad (1)$$

and its ratio,

$$R = \frac{K_I^{min}}{K_I^{max}}. \qquad (2)$$

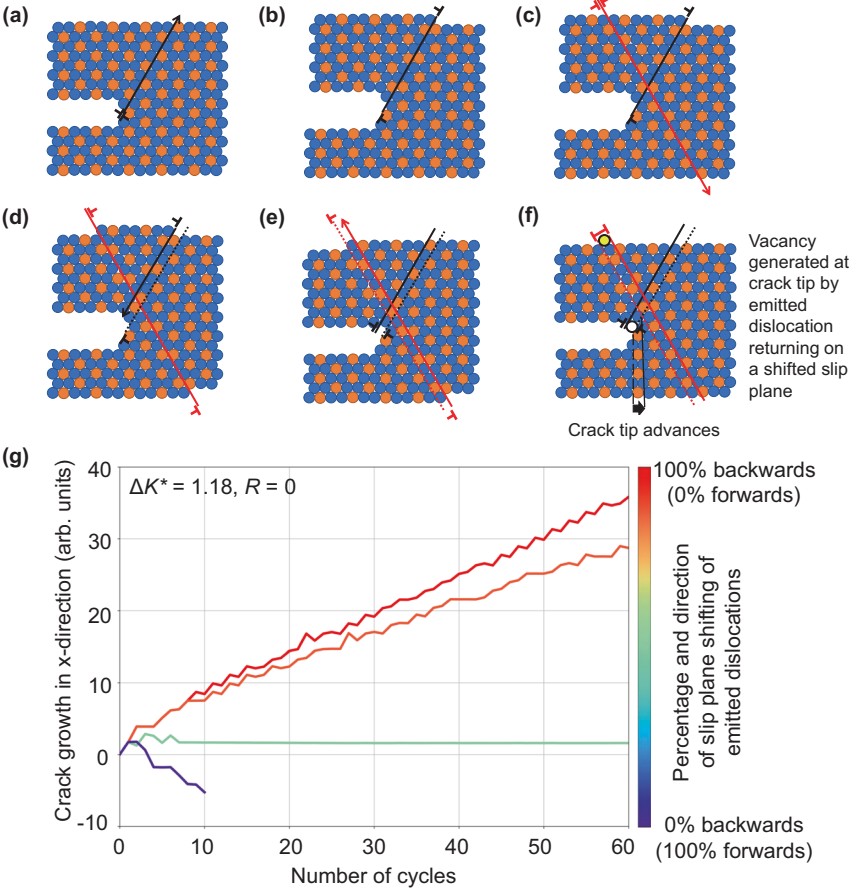

**Fig. 3 Fatigue crack advance with slip trace intersection.** Schematics show the sequence of dislocation nucleation (**a** and **b**), slip trace intersection (**c** and **d**), and the resulting change in slip plane when the dislocation returns and is absorbed on the unload (**e** and **f**). This process leads to the removal of an atom from the crack tip, highlighted in yellow in (**f**), creating a vacancy at the crack tip. Other atom colors are used to show deformation. **g** Crack growth vs. number of cycles with the slip plane of emitted dislocations being changed prior to their absorption into the crack on the unload. Shows that shifting the slip plane of emitted shielding dislocations backwards (in the opposite direction of crack growth) produces sustained crack growth, consistent with the schematic in (**a**)-(**f**). Crack closure was observed in the case when the slip planes were shifted forward, and no crack growth was observed when dislocations were shifted forward and backward in equal proportion. Distances are normalized by the magnitude of the Burgers vector, $b$, and the range of the normalized stress intensity factor that is used to characterize the loading amplitude, $\Delta K^* = \Delta K_I / K_I^{nuc}$.

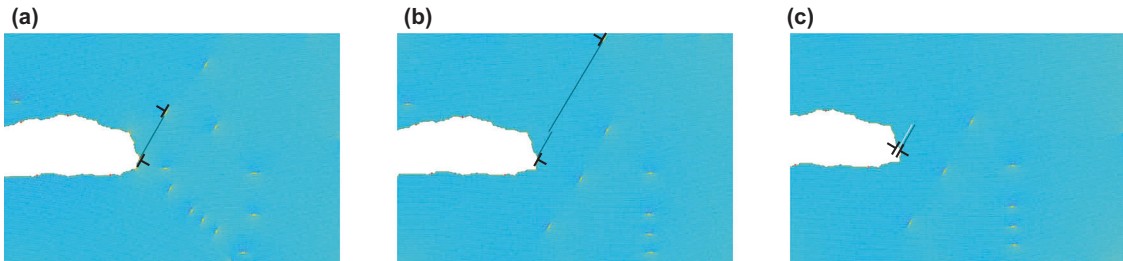

**Fig. 4 Example of crack growth by the intersection of slip traces in a 2D hexagonal lattice simulation, [600$b$ × 300$b$] $\Delta K^* = 3.09$.** The black dislocation dipole in **a** highlights a dislocation that has been emitted from the crack on the upload of cycle 82. Upon further loading, **b** Shows the slip trace of the emitted dislocation (black line) having been intersected by an other dislocation slip trace. Upon unloading, **c** shows the highlighted emitted dislocation having returned to the crack tip in an offset location, leaving a pair of offset slip trace segments and removing an atom from the crack tip. Atoms are colored by strain relative to perfect lattice to illuminate dislocation cores.

In all but the slip plane shifting case, the simulations were performed up to the highest $\Delta K_I$ achievable with our computational resources. Increasing $\Delta K_I$ increases the number of discrete dislocations in the continuum domain, the size of the needed atomistic domain to capture the highly nonlinear zone around the crack tip, and the number of time steps required to capture the deformation per loading cycle. The load ratio, $R$, was selected to balance the computational expensive of simulating large deformations against the sporadic occurrences of crack closure / rewelding that can occur at low $R$ values. Given that our simulations correspond to near threshold fatigue crack growth in vacuum, we do not expect crack closure nor

the value of $R$ to be influential at positive $R$ values[59]. Although, we do note that $R$ independence is not true in other contexts[65].

**3D fcc simulation setup.** The fcc aluminum simulations were performed in a [4960$b$ × 4960$b$] continuum domain with a [77$b$ × 84$b$ × 2$b$] embedded atomistic domain at the crack tip where $b$ is the magnitude of the Burgers vector, which is equivalent to the equilibrium interatomic spacing. The crystal lattice was oriented such that the horizontal and vertical axes were in the directions of [4 7 −1] and

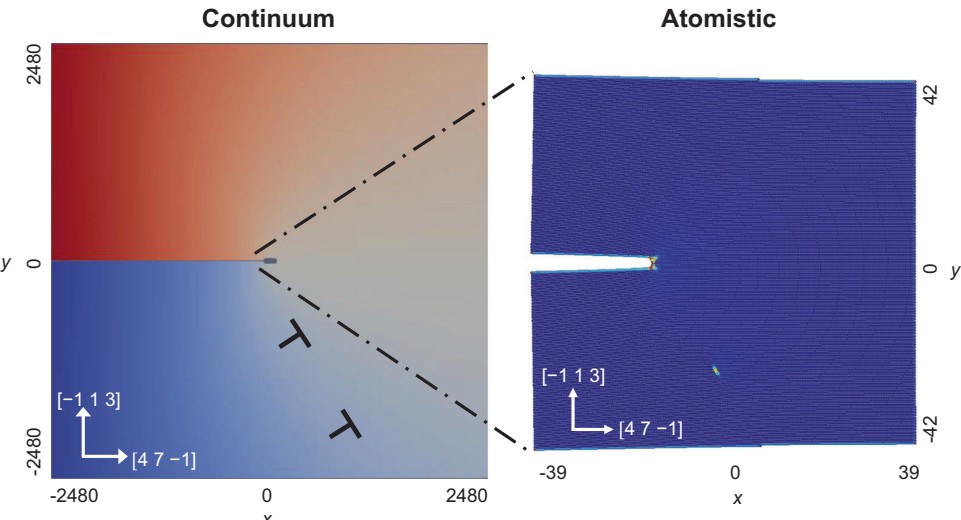

**Fig. 5 Multiscale simulation setup. An atomistic domain (right) is embedded in a linear elastic continuum domain (left).** Dislocations can move between domains on their corresponding slip planes. The lengths given in this figure, which corresponds to the [4 7 −1]/[−1 1 3] crack orientation, are normalized with respect to the magnitude of the Burgers vector, *b*, and the images are not drawn to scale. The continuum coloring represents the macroscopic displacement in the *y* direction and the atomistic coloring corresponds to the potential energy of the atoms, highlighting an edge dislocation that has been emitted from the crack tip.

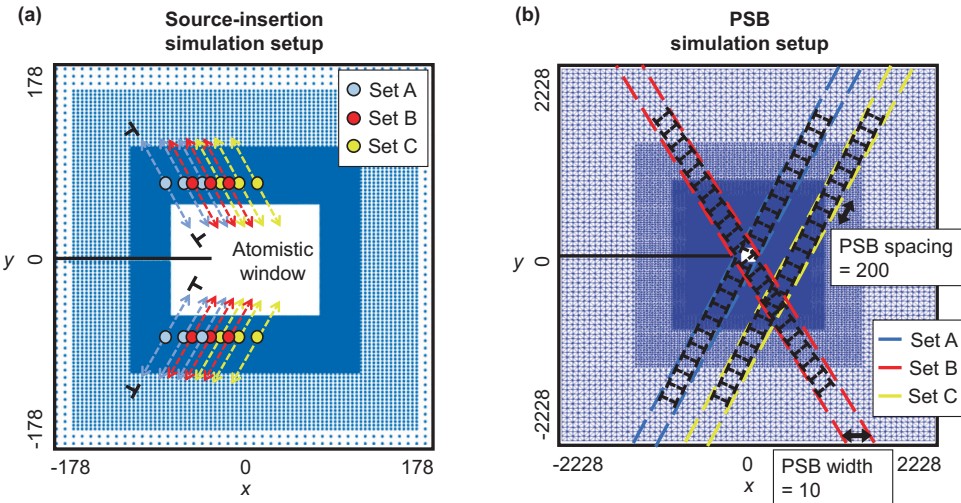

**Fig. 6 Simulation set-ups for source-insertion and persistent slip bands (PSB). a** In each set, an array of three Frank–Read sources was placed in the continuum just outside the atomistic window on each side of the crack. The sources within each array were set apart by 20*b*, and each source emitted a dislocation dipole when its total resolved shear stress exceeded a critical value along the slip plane. The dislocation entering the atomistic window was constructed to be an anti-shielding dislocation gliding towards the crack tip region. **b** In each set, a pre-existing persistent slip band with width of 10*b* and spacing of 200*b* was constructed following the schematics by Sangid[54]. The PSB was arranged to intersect the crack tip region. The lengths given in this figure are normalized with respect to the magnitude of the Burgers vector, *b*, and the images are not drawn to scale.

[−1 1 3], respectively. The primary slip plane is the (1 1 −1), which intersected the crack plane at an angle of 58.5° from the horizontal, and slip consists primarily of edge dislocations in the [0 1 1] direction. The atomistic domain was governed by the Ercolessi and Adams[66] interatomic potential which has been shown to accurately reproduce pure aluminum crack tip behavior[67]. The loading increment after each mechanical equilibration was 0.02 MPa m$^{1/2}$. The $K_I$ value at which the first dislocation nucleates during the initial loading cycle is denoted as $K_I^{nuc}$ and was found to be 0.59 MPa m$^{1/2}$ in the 3D fcc simulations. Given the importance of the dislocation nucleation process for crack advance, we report the value of $\Delta K_I$ normalized by $K_I^{nuc}$, to enable comparison across material systems

$$\Delta K^* = \frac{\Delta K_I}{K_I^{nuc}}. \qquad (3)$$

**2D hexagonal lattice simulation setup**. The 2D hexagonal lattice simulations were performed in a [4464*b* × 4464*b*] continuum domain with an embedded

atomistic domain at the crack tip ranging from [268*b* × 134*b*] to [536*b* × 268*b*] (for simulations without continuum discrete dislocations). The crack was created by removing three consecutive planes of atoms. The 2D hexagonal lattice consists of three slip planes, 60° apart, one of which aligns with the horizontal crack plane. Edge dislocation slip occurs on all three planes.

The atomistic domain is governed by a ductile interatomic pair potential given in ref. [68] (potential A). In the normalized units of the potential, the shear modulus $\mu$ is 10.61 and the critical load for the nucleation of the first dislocation is $K_I^{nuc} = 6.8$. For reference, Griffith cleavage is predicted to occur at $K_I = 7.5$. In the fatigue simulations, the prescribed loading increment after each mechanical equilibration was $K = 0.1$. For discrete dislocations in the continuum, the lattice resistance normalized by the shear modulus was prescribed to be 1.9E−3.

**Source-insertion and PSB models**. The simulation set up for the source-insertion and PSB cases is illustrated in Fig. 6. For source-insertion simulations, an array of three Frank–Read sources, separated by 20*b*, was inserted into the continuum just

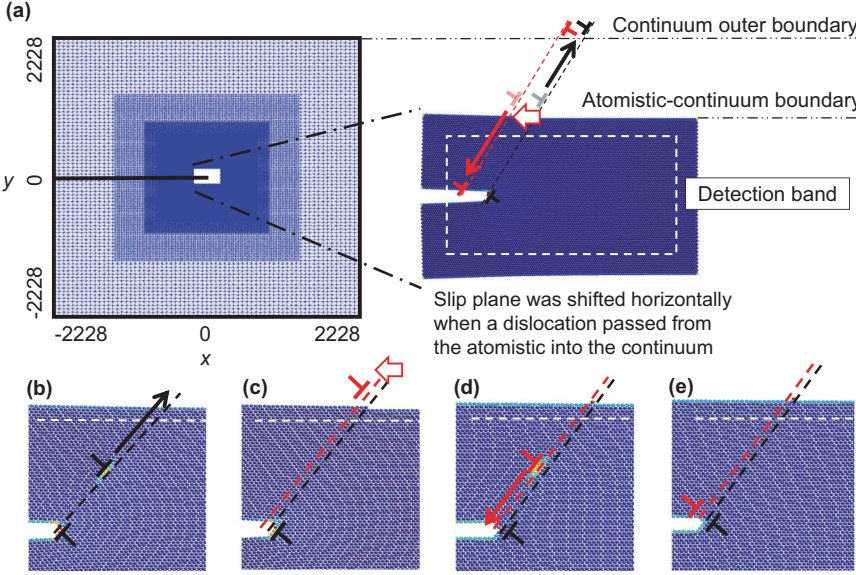

**Fig. 7 Illustration of slip plane shifting implementation.** An overview of the implementation is given in (**a**) that corresponds to the detailed sequence shown in (**b**) through (**e**). When a dislocation (black) nucleated from the crack tip and glided across the detection band inside the atomistic domain (**b**), its slip plane was shifted after being passed across the interface (red), (**c**). In order to maintain the correct displacement fields of both the original and the shifted dislocations, the previously passed dislocation (black) was moved and pinned outside the continuum domain and a new "opposite" dipole dislocation (red) on the shifted slip plane was added. During unload, the shifted (red) dislocation glided back into the atomistic domain, (**d**), and eventually returned to the crack tip (**e**). In all slip plane shifting simulations, the distance of each shift is 1$b$. The lengths given in this figure are normalized with respect to the magnitude of the Burgers vector, $b$, and the images are not drawn to scale.

outside the atomistic window on each side of the crack. When the total resolved shear stress of a source exceeds a critical value as described in ref. [69], a dislocation dipole emits on a slip plane that intersects the crack tip region, as shown in Fig. 6a. A total of three sets of simulations with variations in source array locations were conducted. In each set of the PSB simulations, a pre-existing PSB with width of 10$b$ and spacing of 200$b$ was constructed following the schematics by M.D. Sangid[54] as shown in Fig. 6b. Among the three sets of simulations performed, two of the PSBs were oriented in the same direction as the slip plane making a 60° angle counterclockwise from the $x$-axis, and one 60° clockwise.

**Slip plane shifting model**. Figure 7 conveys the slip plane shifting implementation. After a dislocation emits and reaches the detection band near the atomistic-continuum interface, it is passed from the atomistic domain and placed outside of the continuum domain. At this time, a dislocation dipole is inserted on a neighboring slip plane. One of these dislocations is placed next to the passed dislocation outside the continuum domain and the other is placed just outside of the atomistic domain. Only the dislocation that is placed just outside the atomistic domain is mobile under the action of an applied stress. The slip plane shifting simulations require emitted dislocations to return to the crack during the unloading portion of the loading cycle. To expedite this process for computational efficiency, these simulations had three features different than the simulations presented previously. First, dislocation motion was limited to a zone of 90$b$ from the crack tip. Second, an additional resolved shear stress of 0.12/$\mu$ was added to emitted dislocations in the direction of the crack tip. Third, an $R$ value of zero was chosen. Without these computationally expedient choices, the take-away point of the simulations is expected to be the same, i.e. the change of dislocation plane after emission and before absorption can produce sustained fatigue crack growth. This assertion is supported by the slip plane shifting that naturally occurred during periods of crack growth in the other simulations, e.g. Figs. 2b, c, and 4.

## Data availability
The data presented in the study is entirely the result of computer simulation and can be generated using the shared code.

## Code availability
The code[70] used to generate the data presented in this study is available at https://doi.org/10.5281/zenodo.5735370.

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

## Acknowledgements

This work was supported by the US Office of Naval Research (N00014-17-1-2035 and N00014-20-1-2484). M.Z. received partial support through the Ross-Teteman Fellowship. We thank W.A. Curtin, A.K. Vasudevan, and S. Lynch for discussion.

## Author contributions

D.H.W. developed the concept. W.G. implemented the methodology with the exception of the continuum discrete dislocation component, which was implemented by M.Z. Simulations and analyses involving no continuum discrete dislocations were performed by W.G. The other simulations were performed by M.Z. All of the authors contributed to the paper and approved its publication.

## Competing interests

The authors declare no competing interests.
