## [Peer Review File · Nature Communications]

Title: Atomic mechanism of near threshold fatigue crack growth in vacuumReviewers' comments:

Reviewer #1 (Remarks to the Author):

Comments on paper titled « Atomic mechanism of near threshold fatigue crack growth in vacuum » submitted to Nature Communications by Mingjie Zhao, Wenjia Gu and Derek H. Warner.

The aim of the paper is to investigate which dislocation based mechanism could be at origin of sustained fatigue crack growth in vacuum. The CADD simulation tool combining atomic, dislocation and continuum modelings is used. The authors take advantage of the reduced computation time needed by this code to realize up to 180 fatigue cycles in mode I and for different stress amplitudes with a 2D simulated box as big as $1.2\mu\text{m}$ per side and $2b$ thick. The initial crack length is taken as half the box size. Several dislocation behaviors are tested by either (i) blocking dislocation in the continuum domain so that they cannot provide reversible slip, (ii) introducing dislocation sources emitting dislocations toward the crack and (iii) introducing a preexisting persistent slip band microstructure ahead of the crack. None of the investigated configuration was able to lead to a sustained crack growth, i.e. the crack always get arrested after a few cycles. Then the authors enforced a double cross-slip event on the dislocations emitted from the crack so that they slip back in a parallel plane during the unloading stage. It is found that such a mechanism can indeed sustain the crack growth if the shifted dislocations are located in a backward plane (compared to the crack front position).

The paper is well written and easy to read. The results are new and interesting and can be very useful for the scientific community of damage modeling. The results are nicely supported by a Method additional section where the simulations are better defined.

I have only two minor remarks that could be addressed before publication :

- 1- It would be better to give the unique vector direction $[ijk]$ of the crack configuration in place of the generic $\langle 113 \rangle$ and $\langle 471 \rangle$ directions used in this paper. Changes should be made in the text and more importantly in Figure S1.
- 2- It is not clear how the HCP crystallography is accounted for in the simulation box. More information should be given in the Method supplementary material where the FCC and HCP configurations should be both described.
- 3- Finally the Burgers vectors and slip planes of the involved dislocations should be given.

Reviewer #2 (Remarks to the Author):

The paper presents a very interesting atomistic modeling work on fatigue crack growth in which simulations over large number of cycles could be realized by using a coupled atomistic discrete dislocation (CADD) approach and meeting proper assumptions with respect to size of and slip activities in the atomistic domain. Thereby different mechanisms reported to promote crack growth, in particular slip irreversibility, dislocation sources and persistent slip band (PSB) are investigated. In all cases crack arrest is observed after certain number of cycles accompanied by a reversible/stable dislocation arrangement in the subsequent cycles. However, this is not surprising after knowing that the loading of

the coupled atomistic continuum model is displacement imposed as stated in the methods section. Actually with crack growth the compliance of the model increases what in case of a loading with prescribed displacement results in lower stresses and hence lower stress intensity factors. The diagram on the right side in Figure 1c demonstrates this. Consequently, it is more than obvious that the slip activities will stop after certain number of cycles and the conclusion that the considered mechanisms don't promote sustainable crack propagation is questionable and limited to the simulated loading case. The hypothesis studied afterwards, namely that dislocations emitted from the crack tip have to continuously change their slip plane backwards to cause sustain crack growth, yields different results most probably because the slip plane change as implemented artificially and continuously increases the level of the displacement imposed loading maintaining the slip activities at the crack tip and thus sustaining crack growth. Also the conclusion here that the slip plane change in the right direction must be the mechanism promoting crack growth is questionable as the loading scenario seems to be different from that considered for the other mechanisms (not only in the selected R ratio!). Apart from that the paper is fairly well written and the modeling approach seems to deliver at least consistent interpretable results which however are not sufficient to support the conclusions made. Therefore, the paper cannot be recommended for publication.

Reviewer #3 (Remarks to the Author):

Key results

The authors used simplified modeling assumptions and special crack orientations for atomistic investigation of FCG near threshold. It aims to explain some fundamental challenges with respect to experimental observations for near threshold crack growth. Crack arrest behavior and dislocation interaction is observed and stated as the major indicators for FCG near threshold behavior. One key result (Fig. 3) shows that the necessary condition between continuous growth and crack arrest due to the change of dislocations on the unload.

Validity

The reviewer feels that the proposed method and obtained results offer some interesting supporting evidence of the hypothesis, but the simulation was specially designed and have difficulties to explain some known threshold FCG behavior for its general validity.

- 1) The used small plastic zone size assumption makes the computation tractable but may not be valid for FCG near threshold investigation. For example, small plastic zone size will prohibit the crack closure formation which has been widely observed for many metallic materials. Experimentally observed near threshold behavior is mostly coming from load shedding technique, which inevitably needs the simulation to handle relatively large plastic zone effect.
- 2) Following the above discussion, no simulation showing the crack closure behavior, and this may or may not be true for the reported crack arresting behavior. If crack closure happens, blunting-arresting-resharpening-growing-blunting mechanism will trigger the continuous growth and not the experimentally observed threshold.
- 3) The reviewer also suspect that the proposed method cannot explain the experimentally observed

stress ratio-dependency of near threshold FCG, where the dependency shows significant difference that that in the Paris regime.

4) Threshold FCG approaches the significant stochastic growth regime and may not grow continuously. Sudden jump due to the 3D crack growth (e.g., surface arresting but growing in depth) and other mechanisms will occur. The proposed method is not able to explain the behavior for this well-known phenomenon.

Significance

The topic is of significance for the fatigue and material community. It could potentially benefit the fundamental understanding of FCG near threshold behavior using atomic simulations.

Data and methodology

No experimental data is reported. Simulation methodology appears to be valid as it has already been published elsewhere. The reviewer did not check details of the simulation methodology as it was cited in other peer-reviewed publications.

Analytical approach

NA. This is a simulation study and has not statistical validation tests.

Suggested improvements

It will be great to compare and explain some well-known behavior in near threshold FCG from experimental observations. Also, it may be helpful to suggest some direct validation testing metrics that can support the hypothesis and stated mechanism.

Clarify and context

Very good.

References

Very good.

Response to Reviewers

Reviewer #1 (Remarks to the Author):

The aim of the paper is to investigate which dislocation based mechanism could be at origin of sustained fatigue crack growth in vacuum. The CADD simulation tool combining atomic, dislocation and continuum modelings is used. The authors take advantage of the reduced computation time needed by this code to realize up to 180 fatigue cycles in mode I and for different stress amplitudes with a 2D simulated box as big as $1.2\mu\text{m}$ per side and $2b$ thick. The initial crack length is taken as half the box size. Several dislocation behaviors are tested by either (i) blocking dislocation in the continuum domain so that they cannot provide reversible slip, (ii) introducing dislocation sources emitting dislocations toward the crack and (iii) introducing a preexisting persistent slip band microstructure ahead of the crack. None of the investigated configuration was able to lead to a sustained crack growth, i.e. the crack always get arrested after a few cycles. Then the authors enforced a double cross-slip event on the dislocations emitted from the crack so that they slip back in a parallel plane during the unloading stage. It is found that such a mechanism can indeed sustain the crack growth if the shifted dislocations are located in a backward plane (compared to the crack front position).

The paper is well written and easy to read. The results are new and interesting and can be very useful for the scientific community of damage modeling. The results are nicely supported by a Method additional section where the simulations are better defined.

Authors: We were delighted to read reviewer #1's positive comments on our manuscript. The suggestions for improvement were appreciated and utilized to improve the manuscript.

I have only two minor remarks that could be addressed before publication :

- 1- It would be better to give the unique vector direction $[ijk]$ of the crack configuration in place of the generic $\langle 113 \rangle$ and $\langle 471 \rangle$ directions used in this paper. Changes should be made in the text and more importantly in Figure S1.

Authors: The suggested change has been made to the text and Figure S1 in the revised manuscript.

- 2- It is not clear how the HCP crystallography is accounted for in the simulation box. More information should be given in the Method supplementary material where the FCC and HCP configurations should be both described.

Authors: 3D FCC and 2D hexagonal lattices were presented in the manuscript. It appears the reviewer assumed that a 3D hexagonal lattice was studied. This probably led to his or her point #3. To avoid future readers having such confusion, in the revised manuscript we remind the reader throughout that we are studying a 2D hexagonal lattice by using "2D hexagonal" instead of "hexagonal".

- 3- Finally the Burgers vectors and slip planes of the involved dislocations should be given.

Authors: The burgers vectors and slip planes of the involved dislocations are now given for both the FCC and 2D hexagonal lattices.

“Edge dislocations nucleated from the crack tip glide along the two slip planes 60 degrees from the horizontal axis as shown.”

“This case consists of a ductile 2D hexagonal lattice with edge dislocation slip on 3 planes and glissile dislocation reactions.”

“The crystal lattice was oriented such that the horizontal and vertical axes were in the directions of $[4\ 7\ -1]$ and $[-1\ 1\ 3]$, respectively. The primary slip plane is the $(1\ 1\ -1)$, which intersected the crack plane at an angle of 58.5 degrees from the horizontal, and slip consists primarily of edge dislocations in the $[0\ 1\ 1]$ direction.”

“The 2D hexagonal lattice consists of 3 slip planes, 60 degrees apart, one of which aligns with the horizontal crack plane. Edge dislocation slip occurs on all three planes.”

Reviewer #2 (Remarks to the Author):

Authors: We thank reviewer #2 for critically evaluating our manuscript. We genuinely appreciate the time that was put into the review. The primary concern raised by the reviewer can be attributed to the omission of a key detail in the methods section of the originally submitted manuscript. The revised manuscript addresses this omission.

The paper presents a very interesting atomistic modeling work on fatigue crack growth in which simulations over large number of cycles could be realized by using a coupled atomistic discrete dislocation (CADD) approach and meeting proper assumptions with respect to size of and slip activities in the atomistic domain. Thereby different mechanisms reported to promote crack growth, in particular slip irreversibility, dislocation sources and persistent slip band (PSB) are investigated. In all cases crack arrest is observed after certain number of cycles accompanied by a reversible/stable dislocation arrangement in the subsequent cycles. However, this is not surprising after knowing that the loading of the coupled atomistic continuum model is displacement imposed as stated in the methods section. Actually with crack growth the compliance of the model increases what in case of a loading with prescribed displacement results in lower stresses and hence lower stress intensity factors. The diagram on the right side in Figure 1c demonstrates this. Consequently, it is more than obvious that the slip activities will stop after certain number of cycles and the conclusion that the considered mechanisms don't promote sustainable crack propagation is questionable and limited to the simulated loading case. The hypothesis studied afterwards, namely that dislocations emitted from the crack tip have to continuously change their slip plane backwards to cause sustain crack growth, yields different results most probably because the slip plane change as implemented artificially and continuously increases the level of the displacement imposed loading maintaining the slip activities at the crack tip and thus sustaining crack growth.

Authors: The applied displacements at the outer boundary of the continuum region corresponds to the solution for a sharp crack in an anisotropic linear elastic material subjected to mode I loading.

The reviewer's comment assumes that a fixed crack tip position was used to compute the displacement field. This was not the case. The current location of the crack tip is tracked throughout the simulations using a flood fill algorithm. Accordingly, the applied displacement field evolves over the course of the simulation, avoiding the artifact mentioned by the reviewer. This point is made clear in the revised version of the manuscript with the added text:

“Loads were applied by prescribed displacements at the outer boundary of the continuum region corresponding to the solution for a sharp crack in an anisotropic linear elastic material subjected to mode I loading. The linear elastic solution was calculated using the current position of the crack tip, which was identified using the flood fill algorithm. If the current crack tip position is not tracked and used in the computation of the applied displacements, the stress intensity factor acting on the crack tip would evolve with crack growth.”

Also the conclusion here that the slip plane change in the right direction must be the mechanism promoting crack growth is questionable as the loading scenario seems to be different from that considered for the other mechanisms (not only in the selected R ratio!).

Authors: It is a good point that the difference in R ratio among simulation sets should be mentioned. We have added text to the revised manuscript that addresses the reviewer's point.

“The slip plane shifting simulations require emitted dislocations to return to the crack during the unloading portion of the loading cycle. To expedite this process for computational efficiency, these simulations had three features different than the simulations presented previously. First, dislocation motion was limited to a zone of $90b$ from the crack tip. Second, an additional resolved shear stress of 0.12μ was added to emitted dislocations in the direction of the crack tip. Third, an R value of zero was chosen. Without these computationally expedient choices, the take-away point of the simulations is expected to be the same, i.e. the change of dislocation plane after emission and before absorption can produce sustained fatigue crack growth. This assertion is supported by the slip plane shifting that naturally occurred during periods of crack growth in the other simulations, e.g. Figures 2b, 2c, and 4.”

Apart from that the paper is fairly well written and the modeling approach seems to deliver at least consistent interpretable results which however are not sufficient to support the conclusions made. Therefore, the paper cannot be recommended for publication.

Reviewer #3 (Remarks to the Author):

Key results

The authors used simplified modeling assumptions and special crack orientations for atomistic investigation of FCG near threshold. It aims to explain some fundamental challenges with respect to experimental observations for near threshold crack growth. Crack arrest behavior and dislocation interaction is observed and stated as the major indicators for FCG near threshold behavior. One key result (Fig. 3) shows that the necessary condition between continuous growth and crack arrest due to the change of dislocations on the unload.

Authors: We thank reviewer #3 for his or her time. The overall concern raised by the reviewer is a very good one, i.e. the manuscript should better link to known FCG behavior. We have revised the manuscript towards this goal.

Validity

The reviewer feels that the proposed method and obtained results offer some interesting supporting evidence of the hypothesis, but the simulation was specially designed and have difficulties to explain some known threshold FCG behavior for its general validity.

1) The used small plastic zone size assumption makes the computation tractable but may not be valid for FCG near threshold investigation. For example, small plastic zone size will prohibit the crack closure formation which has been widely observed for many metallic materials. Experimentally observed near threshold behavior is mostly coming from load shedding technique, which inevitably needs the simulation to handle relatively large plastic zone effect.

Authors: Both experiments and discrete dislocation modeling show that yield strength does not by itself influence the FCG threshold, particularly when crack closure is accounted for or does not occur (at high R values or vacuum) as we have simulated. Given that yield strength controls the plastic zone size, one can deduce that plastic zone size does not influence the FCG threshold. Thus, we find it safe to assume that using a reduced plastic zone size in our modeling does not influence the outcome, w.r.t. the reported near threshold FCG behavior. It is a good suggestion to more clearly communicate this line of reasoning in the manuscript. To better communicate this point, the text below has been added.

Regarding the second comment made by the reviewer, the plastic zone size might influence the threshold if load shedding is done too quickly, but this point is not relevant to the manuscript. Specifically, our simulations are performed at constant ΔK and the disconnect between experiment and theory exists even when considering carefully performed experiments where load shedding artifacts are demonstrated to not be occurring.

“This result is consistent with the conclusion of discrete dislocation continuum modeling^{39,40} and experiments^{2,39,41-45} that have shown dislocation glide resistance (and hence plastic zone size) to not directly influence near threshold fatigue crack growth, when crack closure is accounted for or does not occur (at high R values or in vacuum).”

“In other terms, we are assuming that the plastic zone size does not influence near threshold fatigue crack growth behavior, which is consistent with both experiment^{2,39,41-45} and discrete dislocation modeling^{39,40} that show yield strength to not influence the fatigue crack growth behavior, when crack closure is accounted for or does not occur (at high R values or in vacuum).”

2) Following the above discussion, no simulation showing the crack closure behavior, and this may or may not be true for the reported crack arresting behavior. If crack closure happens, blunting-arresting-resharpening-growing-blunting mechanism will trigger the continuous growth and not the experimentally observed threshold.

Authors: The occurrence of crack closure in near threshold fatigue is a long-disputed topic. Nonetheless, we suspect that the majority of experts will agree that closure is not always a governing feature of near threshold FCG. An example is given in our response to point #3, where the experimentally determined FCG threshold value is shown to be independent of R when environmental effects are absent. This case is consistent with the regime that we simulate, i.e. vacuum. Accordingly, the lack of closure in our simulations does not support an argument against their validity. The following text was added to the manuscript to address this and the next point:

“Given that our simulations correspond to near threshold fatigue crack growth in vacuum, we do not expect crack closure nor the value of R to be influential²⁷. Although, we do emphasize that this expectation is not generally true⁶⁹”

3) The reviewer also suspect that the proposed method cannot explain the experimentally observed stress ratio-dependency of near threshold FCG, where the dependency shows significant difference that that in the Paris regime.

Authors: Multiple independent researchers have shown that near threshold FCG is independent of R ratio when the action of the environment is not rate controlling. For example, we show below figure 5 from the 1995 work of Vasudevan and Sadananda. The lack of R dependence in our modeling is consistent with this data, considering that we are modeling FCG in vacuum.

Fig. 5—Experimental data for all types of class I alloys, ΔK_{th} - R plots for all $R > 0$. (a) Composition effects in vacuum, (b) composition effects in environment, and (c) microstructural effects in a 7075 alloy under vacuum.

4) Threshold FCG approaches the significant stochastic growth regime and may not grow continuously. Sudden jump due to the 3D crack growth (e.g., surface arresting but growing in depth) and other mechanisms will occur. The proposed method is not able to explain the behavior for this well-known phenomenon.

Authors: We agree that near threshold FCG may not occur continuously. In fact, our simulation data in figure 1a and 1c is consistent with this point, showing jumps in the crack tip position. In figure 3, where we show sustained crack growth, the crack tip advances continuously, but we assert that this is due to the regular occurrence of slip plane shifting in those simulations. Slip plane

shifting would not be expected to occur so regularly in reality. We have communicated this point in the revised manuscript:

“Further, the near continuous advance of the crack shown in Figure 3b should not necessarily be expected, as slip plane shifting might occur in bursts when it results from dislocation processes in real 3D crystals.”

Significance

The topic is of significance for the fatigue and material community. It could potentially benefit the fundamental understanding of FCG near threshold behavior using atomic simulations.

Data and methodology

No experimental data is reported. Simulation methodology appears to be valid as it has already been published elsewhere. The reviewer did not check details of the simulation methodology as it was cited in other peer-reviewed publications.

Analytical approach

NA. This is a simulation study and has not statistical validation tests.

Suggested improvements

It will be great to compare and explain some well-known behavior in near threshold FCG from experimental observations. Also, it may be helpful to suggest some direct validation testing metrics that can support the hypothesis and stated mechanism.

Authors: We agree. With our above mentioned responses we have tried to better connect to well-known near threshold FCG observations and we have added a statement to the conclusion where we suggest a path for experimental validation of the proposed mechanism.

“An easy path towards experimental confirmation is not obvious, but perhaps tracking individual dislocation slip traces at the crack tip via a marked surface and scanning probe microscopy would be illuminating.”

Clarify and context

Very good.

References

Very good.

REVIEWERS' COMMENTS

Reviewer #1 (Remarks to the Author):

The authors satisfactorily modified the text to account for the suggestions.
I would suggest the authors replace all the occurrence of 'burgers' by 'Burgers' in the text.

Reviewer #2 (Remarks to the Author):

In the revised paper the authors added in the “Methods” section a very important detail concerning the displacement imposed loading applied in their coupled atomistic discrete dislocation (CADD) simulations, namely that the displacements applied were updated during the simulation by tracking the crack tip and calculating them for the current crack length so that the nominal stress intensity factor is kept constant. Hence, the crack arrest observed in the simulations for the mechanisms slip irreversibility, dislocation sources and persistent slip band can no longer be attributed to the reduction of stress intensity factor when the prescribed displacements are not adapted to the compliance increase with crack growth.

However, the main issue remains, namely that the results do not necessarily support the conclusion that the slip plane change is „the“ mechanism for sustained fatigue crack growth in the near threshold regime because of the different R ratio selected for the simulation of this mechanism. The authors argue that the R-ratio was selected in favor of the computational efficiency and no influence of the R-ratio on their results is expected as these results address near threshold crack growth in vacuum which is according to literature independent on R ratio for many materials in the range $R > 0$. Indeed, apart from the fact that the later is not generally true, their simulations need to demonstrate clearly either that the results are independent on the R ratio in the range of the selected R values ($R > 0$) or that the other mechanisms, slip irreversibility, preexisting dislocation sources and persistent slip band would not yield sustained fatigue crack growth with the same loading conditions, low loading amplitude and $R = 0$, selected for the slip plane change mechanism. Otherwise, the above conclusion would over-interpret the results presented so far, so that the publication of the paper even in its revised form cannot be recommended.

Reviewer #3 (Remarks to the Author):

The authors have successfully addressed all my questions.

Reviewer #4 (Remarks to the Author):

I have read the paper in its entirety and I see no fundamental issues with the work. I also find the results thorough and carefully obtained, and find the conclusions compelling and well-supported by the results.

The fundamental conclusion is that of four mechanisms considered, only one (dislocations changing slip planes, which I'll call cross-slip for short) gives rise to sustained crack growth. The specific question remaining seems to be whether or not this overall conclusion can be drawn from the limited set of load ratios, R , considered. On this point, I side with the authors. It is hard to imagine any of the 3 mechanisms that were discounted (irreversibility, sources, PSBs) to be *more* likely to support sustained growth at lower R values, or even different R values, without the additional effects of cross-slip. To require them to produce more, expensive, simulations to demonstrate this seems unnecessary to me.

I should note that this assumes we are talking about $R \geq 0$, to which both the authors (perhaps tacitly) and the other reviewer were confining themselves. Negative values might be another story perhaps, and the authors might want to add a comment to that effect, but this is a small point.

REVIEWERS' COMMENTS

Reviewer #1 (Remarks to the Author):

The authors satisfactorily modified the text to account for the suggestions. I would suggest the authors replace all the occurrence of 'burgers' by 'Burgers' in the text.

Authors: All instances of the word “burgers” have been changed to “Burgers.”

Reviewer #2 (Remarks to the Author):

In the revised paper the authors added in the “Methods” section a very important detail concerning the displacement imposed loading applied in their coupled atomistic discrete dislocation (CADD) simulations, namely that the displacements applied were updated during the simulation by tracking the crack tip and calculating them for the current crack length so that the nominal stress intensity factor is kept constant. Hence, the crack arrest observed in the simulations for the mechanisms slip irreversibility, dislocation sources and persistent slip band can no longer be attributed to the reduction of stress intensity factor when the prescribed displacements are not adapted to the compliance increase with crack growth.

However, the main issue remains, namely that the results do not necessarily support the conclusion that the slip plane change is “the” mechanism for sustained fatigue crack growth in the near threshold regime because of the different R ratio selected for the simulation of this mechanism. The authors argue that the R-ratio was selected in favor of the computational efficiency and no influence of the R-ratio on their results is expected as these results address near threshold crack growth in vacuum which is according to literature independent on R ratio for many materials in the range $R > 0$. Indeed, apart from the fact that the later is not generally true, their simulations need to demonstrate clearly either that the results are independent on the R ratio in the range of the selected R values ($R \geq 0$) or that the other mechanisms, slip irreversibility, preexisting dislocation sources and persistent slip band would not yield sustained fatigue crack growth with the same loading conditions, low loading amplitude and $R = 0$, selected for the slip plane change mechanism. Otherwise, the above conclusion would over-interpret the results presented so far, so that the publication of the paper even in its revised form cannot be recommended.

Authors: We address the reviewer’s concern with respect to differing R ratios at two locations in the text. Further, the text provides a reference that supports our claim of R-ratio independence in vacuum across several experimental data sets.

“The load ratio, $R = K_{\min} / K_{\max}$, was selected to balance the computational expensive of simulating large deformations against the sporadic occurrences of crack closure / rewelding that can occur at low R values. Given that our simulations correspond to near threshold fatigue crack growth in vacuum, we do not expect crack closure nor the value of R to be influential at positive R ratios \cite{Vasudevan1995}. Although, we do mention that R-ratio independence is not true in other contexts \cite{Pippan2016}.”

“The slip plane shifting simulations require emitted dislocations to return to the crack during the unloading portion of the loading cycle. To expedite this process for computational efficiency, these simulations had three features different than the simulations presented previously. First, dislocation motion was limited to a zone of 90° from the crack tip. Second, an additional resolved shear stress of $0.12/\mu$ was added to emitted dislocations in the direction of the crack tip. Third, an R value of zero was chosen. Without these computationally expedient choices, the take-away point of the simulations is expected to be the same, i.e. the change of dislocation plane after emission and before absorption can produce sustained fatigue crack growth. This assertion is supported by the slip plane shifting that naturally occurred during periods of crack growth in the other simulations, e.g. Figures \ref{irreverse_source_PSB}b, \ref{irreverse_source_PSB}c, and \ref{intersection_highcycles}.”

Reviewer #3 (Remarks to the Author):

The authors have successfully addressed all my questions.

Reviewer #4 (Remarks to the Author):

I have read the paper in its entirety and I see no fundamental issues with the work. I also find the results thorough and carefully obtained, and find the conclusions compelling and well-supported by the results.

The fundamental conclusion is that of four mechanisms considered, only one (dislocations changing slip planes, which I'll call cross-slip for short) gives rise to sustained crack growth. The specific question remaining seems to be whether or not this overall conclusion can be drawn from the limited set of load ratios, R, considered. On this point, I side with the authors. It is hard to imagine any of the 3 mechanisms that were discounted (irreversibility, sources, PSBs) to be *more* likely to support sustained growth at lower R values, or even different R values, without the additional effects of cross-slip. To require them to produce more, expensive, simulations to demonstrate this seems unnecessary to me.

I should note that this assumes we are talking about $R \geq 0$, to which both the authors (perhaps tacitly) and the other reviewer were confining themselves. Negative values might be another story perhaps, and the authors might want to add a comment to that effect, but this is a small point.

Authors: We now explicitly state that we are considering only positive R ratios

“at positive R ratios”

and have added a comment on negative R ratios in the final paragraph

“simulations are needed to illuminate the mechanisms controlling environmental, mixed-mode, variable and reversed ($R < 0$) loading effects”